# What Can We Learn about the Bias of Microbiome Studies from Analyzing Data from Mock Communities?

**DOI:** 10.3390/genes13101758

**Published:** 2022-09-28

**Authors:** Mo Li, Robert E. Tyx, Angel J. Rivera, Ni Zhao, Glen A. Satten

**Affiliations:** 1Department of Biostatistics, Bloomberg School of Public Health, Johns Hopkins University, Baltimore, MD 21205, USA; 2Division of Laboratory Sciences, National Center for Environmental Health, Centers for Disease Control and Prevention, Atlanta, GA 30341, USA; 3Department of Gynecology and Obstetrics, School of Medicine, Emory University, Atlanta, GA 30322, USA

**Keywords:** biased measurement, 16s rRNA sequencing, mock community, log-linear model

## Abstract

It is known that data from both 16S and shotgun metagenomics studies are subject to biases that cause the observed relative abundances of taxa to differ from their true values. Model community analyses, in which the relative abundances of all taxa in the sample are known by construction, seem to offer the hope that these biases can be measured. However, it is unclear whether the bias we measure in a mock community analysis is the same as we measure in a sample in which taxa are spiked in at known relative abundance, or if the biases we measure in spike-in samples is the same as the bias we would measure in a real (e.g., biological) sample. Here, we consider these questions in the context of 16S rRNA measurements on three sets of samples: the commercially available Zymo cells model community; the Zymo model community mixed with Swedish Snus, a smokeless tobacco product that is virtually bacteria-free; and a set of commercially available smokeless tobacco products. Each set of samples was subject to four different extraction protocols. The goal of our analysis is to determine whether the patterns of bias observed in each set of samples are the same, i.e., can we learn about the bias in the commercially available smokeless tobacco products by studying the Zymo cells model community?

## 1. Introduction

There is growing appreciation that biases can be introduced into analyses of metagenomic data [1], and that these biases can be introduced in every step of the experiment [2]. Examples of bias that can occur include extraction bias, which favors microbes that are easily lysed, primer-mismatch bias, which favors amplification of DNA from microbes whose sequences match well to primers used in amplification [3,4,5], and DNA amplification biases, which favor amplification of reads with high GC content [6].

There is no universally accepted method to normalize high-throughput microbiome data to achieve bias reduction [7,8]. Unlike other -omics data, it is not always reasonable to assume that only a small number of taxa vary with important variables such as disease status, because microbes form an interacting community. Analyzing model communities’ samples as positive controls has been suggested as a qualitative check on experimental integrity [3,9], yet no full-scale normalization procedure has been developed using model community controls. This is largely because the number of taxa in commercially available model communities is limited, and is orders of magnitude lower than the number of taxa usually seen in experimental data.

One positive step is the recent development of a biological and mathematical model to describe how biases affect the relative frequencies observed in an experiment such 16S rRNA survey [2]. Using this model, it is possible to start describing which procedures and biological conditions increase or decrease bias, using model community data. For example, it is possible to compare several extraction protocols and quantify which protocol has the least bias. 

The motivation for this work stems from surveys of the bacterial content in smokeless tobacco products sold in the US [10]. Cell wall degrading enzymes are commonly used in microbial nucleic acid extraction procedures to obtain more consistent and complete recovery of bacterial nucleic acids. This is particularly useful when disrupting Gram-positive bacteria, generally known to have thicker cell walls [11]. We wondered if the DNA extraction method for tobacco we previously published [12] would be improved by including cell-wall-degrading enzymes to assist the extraction process. We also wondered if the addition of reagents to our protocol, such as RNAProtect^®^ and Lifeguard^®^ that are designed to preserve DNA, might result in a lower overall bias. To answer these questions, we performed 16S rRNA surveys of the ZymoBIOMICS™ microbial community (ZMC) using four methods, corresponding to our original protocol and three modifications; a ‘baseline’ protocol without enzymes, and the baseline protocol with the addition of either enzymes, Lifeguard^®^ or RNAProtect^®^.

Model communities differ from real communities in many important ways. The ZMC we used comprises bacterial cells in liquid suspension. An important question is whether the biases we observe when we analyze the ZMC are the same as we would observe if we measured these biases if these taxa were found in a smokeless tobacco product. Because the taxa in the ZMC are typically not found in smokeless tobacco products, we mixed the ZMC into a biological matrix comprised of Swedish Snus, a smokeless tobacco product that is largely free of microbes because it is pasteurized (i.e., heat treated) [12,13,14,15]. We compared measurements of the biases of each extraction protocol we observed in the ZMC with the biases found in ZMC + Matrix samples. Significant differences in the bias induced by a protocol in ZMC compared to that in ZMC + Matrix would suggest a model community would have to be tailored to the biological matrix that the study samples are found in (e.g., stool, saliva, etc.).

Finally, it is not immediately clear how useful it is to assess bias using a model community having operational taxonomic units (OTUs) that are not present in actual samples, even if a biological matrix is included. Thus, we tried to assess the consistency of the bias findings in our model community data with relative bias information from actual smokeless tobacco samples. Note that without knowing the true OTU frequencies, it is impossible to ascertain bias factors for each method in real samples. However, we can measure the *relative* biases between methods (i.e., the *difference* in log-bias factors between two protocols) and see if the patterns we observe in real samples agree with the patterns observed in model community data. We develop here the statistical methods that allow these comparisons to be made.

All statistical analyses of bias used the approach of Zhao and Satten (2021) [16], which conducts permutation-based inference on a log-linear model of relative frequencies in model communities. This method is implemented in the R package “MicroBias” (available at https://github.com/zhaoni153/MicroBias, accessed on 13 January 2022). All other quantities were calculated in R as well.

## 2. Materials and Methods

### 2.1. Samples and Sample Preparation

This study used data from 4 different types of samples: the Zymo standard comprising cells (ZMC); the Zymo cell standard mixed with a biological matrix (ZMC + Matrix); samples of Swedish Snus, a smokeless tobacco product from Sweden that has been pasteurized; and samples comprising commercially available smokeless tobacco products. 

Samples were processed by four extraction protocols. Protocol 1 (Base) serves as a baseline method and is a modification of a protocol that has been previously described by us [12], but without the use of any additional enzymes. The other three protocols can be considered as modifications of the base method. Protocol 2 (Enzymes) is our original protocol, which here we describe as the base protocol but with added enzymes to facilitate bacterial lysis; to 2.5 mL of bead solution in the extraction tube, we added 62.5 µL of an enzyme cocktail containing 12.5 µL lysozyme (10 mg/mL), 37.5 µL mutanolysin (1 mg/mL) and 12.5 µL lysostaphin (5 mg/mL). We then vortexed briefly and incubated at 37 °C for 30 min before adding 50 µL proteinase K (from a 20 mg/mL stock solution), 0.25 mL of solution SR1 and 0.8 mL of solution SR2, incubating for another 30 min at 55 °C, and then proceeded with the MoBio extraction protocol. Protocol 3 (Lifeguard^®^) followed the baseline protocol, but added Lifeguard^®^ Soil Preservation Reagent (Qiagen, Germantown, MD, USA, formerly MoBio, Carlsbad, CA, USA) prior to extraction, using 1 mL Lifeguard^®^ per 0.5 g tobacco sample. Protocol 4 (RNAProtect^®^) also followed the baseline protocol, but added RNAProtect^®^ Bacterial Reagent (Qiagen, Germantown, MD, USA) prior to extraction, using 1 mL RNAProtect^®^ per 0.5 g tobacco sample. 

The ZMC samples are constructed with a known (theoretical) composition. For the samples we analyzed, the theoretical compositions provided by Zymo based on kit batch number is: *Staphylococcus*—13.3%, *Bacillus*—15.7%, *Lactobacillus*—18.8%, *Listeria*—15.9%, *Enterococcus*—10.4%, *Salmonella*—11.3%, *Escherichia Shigella*—10.0%, *Pseudomonas*—4.6% (Zymo, personal communication). To create the ZMC + Matrix samples, a 75 µL volume of the ZMC solution was pipetted directly onto a 0.5 g snus tobacco sample and vortexed for 30 s. Before continuing with extractions, a 10 min incubation at room temperature was executed to allow complete absorption of ZMC microbes onto the snus substrate, to better simulate a real tobacco product. Sample processing of the tobacco samples is as described in Tyx et al. (2022) [10].

Each sample was analyzed using all four protocols; in addition, we conducted replicate extractions of each sample. For the ZMC, ZMC + Matrix and Swedish Snus data we performed four replicate extractions for each protocol; thus, these sample sets each have 16 observations. The smokeless tobacco samples comprise six different smokeless tobacco products that were purchased in 2016 by Lab Depot (Atlanta, GA, USA) and shipped to CDC [10]. The four extraction protocols were run in triplicate for each of the six products for a total of 6×4×3=72 samples. Because the composition of these samples is unknown, it is impossible to distinguish between a zero count that represents the absence of a taxon from the sample specimen and a zero count that represents undersampling. Since zero counts are problematic in compositional methods, we restrict our analyses to the 10 taxa that were detected in all 12 samples of at least one of the six tobacco products. The specific taxa can be found in Appendix A.

All replicates from all four sample sets were then subject to 16S rRNA sequencing as described in [10]. We did not observe any significant or consistent trends in recovered DNA using the various modifications of our extraction process; for additional information see Appendix A. After DNA was extracted from the products, the V4-V5 region of the 16S rDNA was amplified. Data was imported into a custom R pipeline, using the ‘phyloseq’ [17], ‘dada2’ [18], and ‘MicroBias’ [16] R packages for analysis. Observed communities were overall consistent among replicates and extraction methods within all samples (Appendix A).

### 2.2. Statistical Methods

All inference on bias is based on the model of Zhao and Satten (2021) [16], which extends the model of McLaren et al. (2019) [2], and which posits that the observed relative abundance of the jth taxon (out of *J* total taxa) in the ith sample (${\tilde{p}}_{ij}$) is related to true taxon relative abundances in the ith sample ($p_{ij})$ by
(1)$$E\left({\tilde{p}}_{ij}\right)=\frac{e^{\gamma_{ij}}p_{ij}}{\sum_{j^{\prime }=1}^{J}e^{\gamma_{ij’}}p_{ij’}}\;,$$
where the *log-bias factor* $\gamma_{ij}$ for the *j*th taxon in the *i*th sample is given by $\gamma_{ij}=X_{i\cdot }\;\beta_{\cdot j}$ and where $X_{i\cdot }$ is the ith row of a N×M design matrix X containing covariate values for the ith sample, $\beta_{\cdot j}$ is the jth column of a M×J matrix of regression coefficients β. Here, by taxon we mean either operational taxonomic unit (OTU) or amplicon sequence variant (ASV). 

Model (1) is equivalent to the log-linear compositional model
(2)$${\tilde{Y}}_{ij}-Y_{ij}=X_{i\cdot }\;{\tilde{\beta }}_{\cdot j}+\epsilon_{ij},$$
where ${\tilde{Y}}_{ij}=\ln{\tilde{p}}_{ij}-\overset{J}{\underset{j^{\prime }=1}{\sum }}\ln{\tilde{p}}_{ij’}$ and $Y_{ij}=\ln p_{ij}-\sum_{j^{\prime }=1}^{J}\ln p_{ij’}$ are the centered log-ratios of the observed and true relative abundances, respectively, and $\epsilon_{ij}$ is mean-zero random error; ${\tilde{\beta }}_{ij}$ is related to *β* by ${\tilde{\beta }}_{ij}=(\beta_{ij}-{\bar{\beta }}_{i\cdot })$ where ${\bar{\beta }}_{i\cdot }=\overset{J}{\underset{j^{\prime }=1}{\sum }}\beta_{ij’}$ is the mean of the *i*th row of *β*. Inference on elements of the matrix of parameters $\tilde{\beta }$ is performed by using least squares and *p*-values are calculated using permutation. We let $\hat{\beta }$ denote the least-squares estimator of $\tilde{\beta }$. In the simplest case considered here, $X_{ij}=\left(I\left[m_{i}=1\right],I\left[m_{i}=2\right],I\left[m_{i}=3\right],I\left[m_{i}=4\right]\right)$ where $m_{i}$ is the protocol used for the *i*th sample and $I\left[m_{i}=k\right]=1$ if $m_{i}=k$ and 0 otherwise. For this model, the bias factors for taxon *j* for a sample using protocol *k* are ${\tilde{\beta }}_{k\cdot }$ (the *k*th row of $\tilde{\beta }$). Inference is carried out by calculating an *F*-statistic that compares the residual sum of squares for an unrestricted model to the residual sum of squares calculated under the restriction of a null hypothesis such as ${\tilde{\beta }}_{k1}=\cdot \cdot \cdot ={\tilde{\beta }}_{kJ}=0$ for some fixed *k* (which in this example corresponds to the hypothesis of no bias for protocol *k*). For all log-linear models, 10,000 permutations were used to evaluate statistical significance. We can estimate the effect size of the bias of protocol *k* by $\frac{1}{J}\overset{J}{\underset{j=1}{\sum }}{\hat{\beta }}_{kj}^{2}$. Additional information on how model covariates were coded and effect sizes were measured for the models used here can be found in Section A.1.

When two sample sets do not have the same taxa, the approach based on log-linear models may fail. To handle this case, we note that (2) also implies a relationship between the Aitchison distance [19] between samples and estimates $\hat{\beta }$ of the regression coefficients $\tilde{\beta }$. In particular, if we calculate the Aitchison distance $D_{kk’}$ between the mean log-ratio ${\tilde{Y}}_{i\cdot }$ for all samples processed using one protocol (say, protocol *k*) and the mean log-ratio for all samples processed using a second protocol (say, protocol k’), then $D_{kk’}^{2}=\overset{J}{\underset{j=1}{\sum }}{\left({\hat{\beta }}_{k,j}-{\hat{\beta }}_{k^{\prime },j}\right)}^{2}$, which is the (squared) Euclidean distance between the estimated vector of log-bias factors for protocols *k* and k’. 

The relationship between the Aitchison distance between samples and the Euclidean distance between bias factors implies that a second way to determine the relationship between two extraction protocols in different sample sets is to compare the Euclidean distance matrices of the bias factors, across the two sample sets. In essence, these tests whether the relationships among the protocols (e.g., whether Base is closer to Enzymes, RNAProtect^®^ closer to Lifeguard^®^, etc.) is the same in the two sample sets. There are a number of tests of this type, the Mantel test [20] being the oldest and best known. The Mantel test essentially calculates the correlation between the elements of the two distance matrices (after they are vectorized, i.e., converted into vectors). Here, we use a test based on the generalized RV-coefficient [21] proposed by Minas et al. (2013) [22] that has been shown to have more power than the Mantel test. Importantly, this approach can be used with only the *relative* bias factors that are available even when true relative abundances are not known.

When true taxa relative abundances are not known, it is still possible to use (2) to ascertain the *relative* bias of each method, i.e., the difference bias factors between any two protocols. This is accomplished by analyzing *pairs* of samples i and i’, where sample i (i’) was process by protocol k (k’). If we subtract a version of Equation (2) for observation i’ from the version for observation i while noting the true relative frequencies are the same for all observations, we find
(3)$${\tilde{Y}}_{ij}-{\tilde{Y}}_{i^{\prime }j}=(X_{i\cdot }-X_{i^{\prime }\cdot })\;{\tilde{\beta }}_{\cdot j}+e_{i,i^{\prime },j},$$
which resembles (2) but with the centered log-ratio of the relative abundances of observation i’ replacing the centered log-ratio of the true relative abundances. Note that only *differences* between rows of $\tilde{\beta }$ are identifiable for the design matrix $X_{ij}$ we use here, as the term $(X_{i\cdot }-X_{i^{\prime }\cdot })\;{\tilde{\beta }}_{\cdot j}$ always corresponds to ${\tilde{\beta }}_{k,j}-{\tilde{\beta }}_{k^{\prime },j}$. Thus, one row of $\tilde{\beta }$ is not identifiable. For simplicity, we take the last (fourth) row to correspond to the ‘reference’ protocol against which the bias of all other protocols is measured, and similarly remove the last (fourth) element of $X_{i\cdot }$ when fitting (3). When calculating the Euclidean distance between the rows of $\hat{\beta }$ it is sufficient to add a row for the ‘reference’ protocol that has all zero values, as the Euclidean distance is already a function of the differences between rows of $\hat{\beta }$. Alternatively, we may column-center the matrix $\hat{\beta }$ as this does not affect the Euclidean distances $D_{kk’}^{2}=\overset{J}{\underset{j=1}{\sum }}{\left({\hat{\beta }}_{k,j}-{\hat{\beta }}_{k^{\prime },j}\right)}^{2}$, but has the salutary effect that the Gower-centered (squared) Euclidean distance needed for the GRV statistic is then $\hat{\beta }{\hat{\beta }}^{T}$. Because inference using (3) is based on permutation, there is no need to adjust the *F*-statistic to account for the dependence introduced into the data by using pairs of observations.

Inference for the Mantel and GRV tests is usually based on permutation of rows and columns of one of the distance matrices [20,22]. Here, we included the additional variability due to sampling by first drawing a bootstrap sample of replicates (preserving the total number of replicates for each protocol) for each sample set, recalculating $\hat{\beta }$ and recalculating the Euclidean distance between the rows of $\hat{\beta }$ to yield a bootstrap replicate of the distance matrices *D* for each sample set. We then permuted the rows and columns of one of the distance matrices and recalculated the GRV statistic. We used the distribution of GRV test statistics obtained in this as the null distribution and compared the observed GRV statistic to this null distribution to assess significance. Note that for tests of this kind, the null hypothesis is that there is no correlation between the distance matrices, which here implies that the distance relationships between the four protocols are unrelated. The empirical one-sided *p*-values are simply the fractions of permuted test statistics that are greater than the statistic for the observed data. All *p*-values for GRV statistics presented here are based on 500,000 permutations.

## 3. Results

### 3.1. Observed Counts in the Swedish Snus Matrix

We examined the Swedish Snus samples to confirm that counts for the eight taxa in the Zymo model community were small. In almost all cases, we observed only negligible counts for these taxa by any of the four protocols. The sole exception was that we observed substantial counts for *Pseudomonas* in all four Swedish Snus replicates that were extracted using the Enzymes protocol. Excluding these four values, of the remaining 108 values, 70 were zero, and 105 were less than 100. Of particular note is that none of the three other protocols detected *any* counts for *Pseudomonas* for any replicate. There are two possible explanations for these findings: First, *Pseudomonas* is actually present in Swedish Snus, but the bias factors for *Pseudomonas* in the three protocols where it was not detected are all negative infinity; or that the observed *Pseudomonas* counts in the Swedish Snus samples extracted using enzymes are spurious, the result of either contamination or sample mix-up. Although we favor this second conclusion, we have repeated all analyses involving ZMC + Matrix excluding *Pseudomonas*. These results are available in the Appendix A, and do not materially differ from the results we present here that include *Pseudomonas.*

### 3.2. Comparing the Bias of Extraction Methods in Model Community Data

We first tested for evidence of bias in each of the four protocols. We found strong evidence that each method introduced bias into the observed relative abundances; these results are shown in the upper half of Table 1. The estimated bias of the Lifeguard^®^ protocol was the largest (0.194), although the *p*-value for RNAProtect^®^ was the smallest ($p<10^{-4}$). The effect sizes of the bias for each protocol are quite similar, ranging from 0.135 to 0.194. The *p*-value for an overall test of the presence of bias was $p<10^{-4}$ with effect size 0.159. Table 1 also shows the estimated log-bias factors $\hat{\beta }$ for each protocol. Excluding *Staphylococcus*, *Enterococcus* and *Lactobacillus*, which have relatively small log-bias factors compared to the other taxa, there is substantial agreement among the log-bias factors across protocols. In the lower triangle of Table 2, we show the results of testing whether each pair of protocols have the same log-bias factors, as well as the result of an overall test of equality of all the log-bias factors. The effect sizes and associated *p*-values for testing each pair of protocols are listed in related cells. These results indicate when the four protocols are applied to the ZMC sample set they all produce a similar bias. The *p*-value for testing the difference of the bias produced by all four protocols is 0.165, further supporting the claim that each protocol generates a similar bias. Additional information on how variables were coded and how effect size was measured is given in the Section A.1. Versions of Table 1 and Table 2 calculated excluding *Pseudomonas* can be found in Appendix A.

### 3.3. Effect of Tobacco Matrix on Bias in Model Community Data

Results of our analyses of the ZMC + Matrix samples can also be found in Table 1 and Table 2. As with the ZMC samples, there is substantial evidence that each protocol induces a bias. Unlike the ZMC samples, there is substantial evidence of heterogeneity of bias, with estimates of the extent of bias ranging from 0.096 (enzymes) to 0.287 (base). Further, except for base and Lifeguard^®^, the overall concordance in the bias factors for the four protocols seen in the ZMC samples is absent in the presence of the Snus matrix. This is also apparent in Table 2, where the upper triangle gives *p*-values for the results of tests of the equivalence of bias between pairs of protocols. Now only base and Lifeguard^®^ show no evidence of a difference in bias.

In Table 3 we show the results of tests of whether the bias in the ZMC samples is the same as that in the ZMC + Matrix samples. Except for the enzymes protocol, the biases are significantly different with or without the Snus matrix. Additional information on how variables were coded and how effect size was measured is given in the Section A.2. Results calculated without *Pseudomonas* can be found in Appendix A.

### 3.4. Comparing Bias in ZMC and in ZMC + Matrix Samples to Bias in Real Samples

Because there are only three taxa in common between the Zymo community and the 10 Tobacco taxa we analyzed (*Bacillus*, *Staphylococcus* and *Lactobacillus*), and because the true relative abundances of any taxa in the smokeless tobacco products is unknown, we used the distance-based approach to compare the behavior of the four protocols in the ZMC, ZMC + Matrix and tobacco samples. We used the pairwise approach in Equation (3) to analyze these samples. Because the six tobacco products can be expected to have different true relative abundances, we restricted our analyses to only use pairs of samples from the same product.

Table 4 shows the pairwise Euclidean distances between the (relative) bias estimates for each pair of protocols, for the ZMC, ZMC + Matrix and tobacco samples. The distances have been scaled by their matrix (Frobenius) norm to facilitate comparison. A version of Table 4 in which distances for the Zymo sample sets are calculated without *Pseudomonas* is available in Appendix A.

The *p-*value for the GRV test obtained by our permutation analysis comparing ZMC + Matrix and tobacco samples was *p =* 0.0588, which is borderline significant (at the α = 0.05 level). Excluding *Pseudomonas*, the *p-*value was *p =* 0.0342. These *p-*values indicate substantial agreement between the patterns of bias found in the ZMC + Matrix and tobacco samples. 

To develop confidence in these distance-based measures, we also revisited the results we previously obtained using the log-linear model (2), retesting some of these hypotheses using the distance-based approach. We found that the *p-*value for the GRV test comparing ZMC samples and the real tobacco samples was p = 0.5356 (*p =* 0.5020 when *Pseudomonas* is excluded), indicating very little evidence of any concordance between the behavior of the four protocols in ZMC and tobacco samples. Similarly, we retested the comparison of ZMC samples and the ZMC + Mock samples, and found *p =* 0.4345 (*p =* 0.4040 when *Pseudomonas* is excluded), indicating no agreement between bias patterns in ZMC and ZMC + Mock samples, which again agrees with the results obtained using the log-linear model (2). 

## 4. Discussion

In this paper we have examined the use of model community data to make inferences about which laboratory protocols have the least amount of bias. Further, we have examined whether conclusions reached by using a model community that is just a mixture of cells will be different from conclusions reached by using a model community that embeds a known quantity of cells in a biological matrix. We found that, although the bias of all four protocols was quite similar in the ZMC samples, only one protocol behaved similarly when the Zymo cell solution was mixed with a Swedish Snus biological matrix. Although this is only a single example, it suggests that model communities should be embedded in a biological matrix that is appropriate for each sample type (e.g., stool, saliva) of possible interest.

We next addressed the question of whether a model community, when embedded in a biological matrix, shows bias patterns that are similar to the bias patterns we observe in real biological (in this case, smokeless tobacco product) samples. Here, we show there is borderline evidence that the two sample sets show the same patterns of bias, which gives hope that a well-constructed model community (e.g., within an appropriate biological matrix) would allow us to reach conclusions about bias that would generalize to real biological samples. Finally, we can conclude that the original protocol from Tyx et al. (2016) [12], the protocol with enzymes, appears to perform best, having the lowest bias in the ZMC + Matrix samples and the most similar behavior when comparing biases in the ZMC with and without the biological matrix.

At present, there has been no systematic examination of the sources of variability in determining bias factors. We can expect that at least some of the differences in bias factors can be explained by variables such as Gram classification or a measure of primer mismatch. This could be an important and useful undertaking until experimental methods improve enough that bias becomes negligible. 

In addition to determining which extraction protocol provides the least biased relative abundance measurements in smokeless tobacco products, we hope this work can provide a roadmap for others who want to study the sources and magnitudes of biases in their microbiome protocols. In particular, the ability to draw conclusions, however limited, on the similarity of the performance of protocols in a model community and in real samples is encouraging. Although we have considered here only extraction protocols, the methods we provide are suitable for any type of bias that can arise in microbiome studies. By comparing the (scaled) distance between protocols in two types of samples using the generalized RV coefficient, we are able to draw conclusions on whether these protocols behave similarly, even if we do not know the true relative abundances of any of the taxa. It is important, however, to recognize an implicit assumption that the sets of taxa present in the two sets of samples are not confounded by variables that affect bias. For example, if the proportion of Gram-positive taxa in the Zymo model community were substantially different from the proportion of Gram-positive taxa used in our analyses of the smokeless tobacco products, then some of the dissimilarity in the distances matrices may be explained by the set of taxa we used, rather than differences in the behavior of the protocols. If these potentially confounding factors were identified, weighted distances could be used to ensure the distribution of taxon-level confounders was the same in both sets of samples.

## Figures and Tables

**Table 1 genes-13-01758-t001:** Tests and estimates of log-Bias Factors ${\tilde{\beta }}_{j,k}$ for taxa in the ZMC and ZMC + Matrix samples.

	Taxon	Base	Enzymes	Lifeguard^®^	RNAProtect
	ZMC				
Estimated $\hat{\beta }$	*Bacillus*	0.744	0.530	0.751	0.872
	*Listeria*	0.349	0.534	0.383	0.286
	*Staphylococcus*	−0.006	0.136	0.142	−0.201
	*Enterococcus*	0.082	−0.051	0.140	0.058
	*Lactobacillus*	−0.023	0.053	0.099	−0.221
	*Escherichia/Shigella*	−0.399	−0.433	−0.572	−0.302
	*Salmonella*	−0.414	−0.450	−0.578	−0.318
	*Pseudomonas*	−0.333	−0.318	−0.365	−0.174
Effect Size		0.141	0.135	0.194	0.145
Test of Presence of Bias		0.0021	0.0017	0.0003	<1 × 10^−4^
	ZMC + Matrix				
Estimated $\hat{\beta }$	*Bacillus*	0.730	0.013	0.842	0.735
	*Listeria*	0.487	0.395	0.448	0.319
	*Staphylococcus*	−0.282	0.315	−0.215	−0.738
	*Enterococcus*	0.414	−0.408	0.281	0.020
	*Lactobacillus*	0.438	0.402	0.340	0.253
	*Escherichia/Shigella*	−0.632	−0.285	−0.590	−0.233
	*Salmonella*	−0.665	−0.292	−0.621	−0.249
	*Pseudomonas*	−0.490	−0.140	−0.485	−0.107
Effect Size		0.287	0.096	0.265	0.172
Test of Presence of Bias		<1 × 10^−4^	0.0002	<1 × 10^−4^	<1 × 10^−4^

**Table 2 genes-13-01758-t002:** Effect sizes and *p-*values for tests of pairwise bias differences for four extraction protocols in ZMC (lower triangle) and ZMC + Matrix (upper triangle) samples.

		ZMC + Matrix (Overall: 0.152 (*p* < 1 × 10^−4^))			
**ZMC (Overall: 0.029** **(p = 0.1646))**		Base	Enzymes	Lifeguard^®^	RNAProtect
	Base		0.242 (p < 1 × 10^−4^)	0.006 (p = 0.4384)	0.113 (p = 0.0006)
	Enzymes	0.016 (p = 0.4212)		0.221 (p < 1 × 10^−4^)	0.231 (p < 1 × 10^−4^)
	Lifeguard^®^	0.012 (p = 0.5032)	0.019 (p = 0.3092)		0.098 (p = 0.0016)
	RNAProtect	0.018 (p = 0.3907)	0.054 (p = 0.0738)	0.053 (p = 0.0797)	

**Table 3 genes-13-01758-t003:** Tests of effect of Snus Matrix on bias parameters for four extraction protocols.

	*p*-Value (10,000 Permutation Replicates)				
	Base	Enzymes	Lifeguard	RNAProtect	Overall
Effect Size	0.070	0.081	0.029	0.069	0.062
*p*-value	0.0134	0.0130	0.2126	0.0118	0.0032

**Table 4 genes-13-01758-t004:** Scaled pairwise Euclidean distances (divided by their Frobenius norms) between four extraction protocols for ZMC, ZMC + Matrix and tobacco samples.

Protocol Pairs	ZMC	ZMC + Matrix	Tobacco
Base vs. Enzymes	0.2141	0.3645	0.3564
Base vs. Lifeguard	0.1897	0.0583	0.1776
Base vs. RNAProtect	0.2270	0.2490	0.1830
Enzymes vs. Lifeguard	0.2319	0.3483	0.3769
Enzymes vs. RNAProtect	0.3970	0.3558	0.3700
Lifeguard vs. RNAProtect	0.3940	0.2321	0.1704

## Data Availability

Sequences were uploaded to National Center for Biotechnology Information (NCBI) Sequence Read Archive (SRA), accession # PRJNA507122. Data used in these analyses has been uploaded to https://github.com/zhaoni153/tobaccoMicrobiomeData (accessed on 27 September 2022).

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
