# Peer review of "What Can We Learn about the Bias of Microbiome Studies from Analyzing Data from Mock Communities?"

_genes, 2022, doi:10.3390/genes13101758_

Round 1
Reviewer 1 Report
The study was mainly about detecting bias in microbiome studies caused by extraction methods, and the results were not unexpected. The statistics developed in this study could be applied to estimate other type of bias in microbiome research, and this is of interest to the society. There is no technique issues identified in this study, and some minor points could be better explained as below:
1) Please provide rationals or references for the "10 minutes incubation" before sampling (line 112). This is very important, especially for mix ZMC samples with Swedish Snus.
2) Please provide data to support or explain with detail about "largely bacteria-free" (line 63)
3) line 48-50, maybe it is obvious to researchers doing microbiome, please explain concisely why "cell-wall degrading enzyme" matters here (lines48-50)
Reviewer 2 Report
The study used model bacterial communities to investigate any bias between experimental protocols in microbial analysis. In one protocol. There was variation. These experiments were repeated in different biological matrices. Two sample sets showed similar biases. Some conclusions about bias source were discussed to be variables such as Gram classification or a measure of primer mis-match. Statistical analysis of bias was estimated by permutation-based inference on a log-linear model of relative frequencies in model communities. I think this is an original contribution systematically investigating variability sources in bias of microbial communities. The experimental results were presented and discussed in detail.
Author Response
We thank the reviewer for the nice comments.